# Predicted Bacterial Metabolic Landscapes of the Sumaco Volcano: A Picrust2 Analysis of 16S rRNA Data from Amazonian Ecuador

**DOI:** 10.3390/microorganisms14010094

**Published:** 2026-01-01

**Authors:** Pablo Jarrín-V, Julio C. Carrión-Olmedo, Pamela Loján, Daniela Reyes-Barriga, María Lara, Andrés Oña, Cristian Quiroz-Moreno, Pablo Castillejo, Gabriela N. Tenea, Magdalena Díaz, Pablo Monfort-Lanzas, C. Alfonso Molina

**Affiliations:** 1Instituto Nacional de Biodiversidad, Quito 170506, Ecuador; julio.carrion@biodiversidad.gob.ec (J.C.C.-O.); mlaragalarza@gmail.com (M.L.); angelandres.ona@gmail.com (A.O.); 2Department of Horticulture and Crop Science, The Ohio State University, Columbus, OH 43210, USA; 3Grupo de Investigación en Biodiversidad, Medio Ambiente y Salud (BIOMAS), Universidad de Las Américas, Quito 170530, Ecuador; 4Biofood and Nutraceutics Research and Development Group, Faculty of Engineering in Agricultural and Environmental Sciences, Universidad Técnica del Norte, Ibarra 100150, Ecuador; gntenea@utn.edu.ec; 5Facultad de Ingeniería Química, Instituto de Investigación en Zoonosis-CIZ, Universidad Central del Ecuador, Quito 170129, Ecuador; 6Institute of Integrative Systems Biology (I2SysBio), University of Valencia and CSIC, 46010 Valencia, Spain; pablo.monfort@i-med.ac.at; 7Institute of Medical Biochemistry, Biocenter, Medical University Innsbruck, 6020 Innsbruck, Austria; 8Institute of Bioinformatics, Biocenter, Medical University Innsbruck, 6020 Innsbruck, Austria; 9Facultad Medicina Veterinaria y Zootecnia, Instituto de Investigación en Zoonosis-CIZ, Universidad Central del Ecuador, Quito 170129, Ecuador; alf.molina7@gmail.com

**Keywords:** Sumaco, microbiome, metabolism, altitude, sulfur, PICRUSt2, adaptation

## Abstract

The Sumaco volcano in Ecuador, which has a distinct geological origin from the Andes and is located in the Amazon basin, offers an opportunity to study untouched microbiomes. We explored comparative patterns of abundance from predicted functional profiling in soil samples collected along the elevation and sulfur gradients on its slopes. Using 16S rRNA gene metabarcoding, we inferred metagenome functional profiles, contrasting sample groups by altitude or soil sulfur concentration. We inferred that high-altitude communities may have higher predicted abundance for anaerobic metabolism (crotonate fermentation), coenzyme B12 synthesis, and degradation of diverse carbon sources (sugars and octane). High-sulfur soils were associated with an inferred enrichment of pathways for degrading complex organic compounds and nitrogen metabolism, reflecting adaptation to unique geochemical conditions. In contrast, low-sulfur soils are consistent with a higher predicted abundance of glycerol degradation. Within the limitation imposed by the potential weak associations of the applied predicted functional profiling to actual gene content, we propose that the inferred metabolic changes represent different ecological strategies for resource acquisition, energy generation, and stress tolerance, and they are optimized for varying conditions in this unique volcanic ecosystem. Our findings highlight how environmental gradients shape soil microbiome functional diversity and offer insights into microbial adaptation in Sumaco’s exceptional geochemistry within the Amazon. Further efforts linking functional predictions back to specific taxa will offer a complete ecological perspective of the microbiome exploration in the Sumaco volcano.

## 1. Introduction

Volcano slopes are characterized by pronounced environmental gradients that change with altitude. These include variations in temperature, precipitation, solar radiation intensity, and the types of vegetation that can thrive at different elevations [1]. A previous study by the Ecuadorian Microbiome Project EcuMP analyzed the influence of elevation and associated soil physicochemical parameters on the microbial richness and community structure of the Sumaco Volcano [2]. Our study, the first to assess the microbiome of an Amazonian volcano, provides an initial look at the effect of sulfur as a significant element that explains bacterial community structure; this was particularly relevant, as neither elevation nor pH showed stronger explanatory power over sulfur [2]. Sulfur was also not significantly correlated with either elevation or pH [2]. The 16S rRNA gene amplicon dataset recovered from the Sumaco Volcano expedition constitutes a critical resource for advancing the microbial ecological characterization and functional inference of this underexplored Amazon soil environment.

Given the unique geological and ecological characteristics associated with the immaculate forests cradled by the Sumaco Volcano [3,4], the study of undisturbed microbiomes is an opportunity to understand the properties and characteristics of their functional diversity and metabolic potential. Knowledge about this pristine soil environment will increase its value as a standard against which anthropically degraded soil environments could be compared and as a source of potential applications in bioremediation, ecosystem restoration, and climate change assessment [5,6,7].

The Amazon forest is no exception to the increasing deterioration of the natural capital of the Earth [8]. The forest soils are affected by a growing agricultural frontier [9]. This change alters the physicochemical conditions of the soil; modifies the natural structure and diversity of the microbial communities; and transforms the pool of the functional genes, networks of microbial metabolism, and functional properties of microbiomes [7,10,11,12]. Thus, assessments of soil functional value and biological diversity are valuable because soil is a diminishing and deteriorating natural resource [13].

Although elevation gradients influence biodiversity distribution, abundance, and structure, there is no all-encompassing pattern or mode by which elevation acts on determinate groups of living organisms [7,14]. Syntheses and meta-analyses have not been able to provide unequivocal patterns by which elevation determines or correlates with microbial diversity and community composition [15,16,17,18]. For the Neotropics, a decrease in microbial richness has previously been observed at higher altitudes along an elevational gradient from the Amazon to the Andes [19], from glacial elevations along a gradient in the Cayambe volcano [20], and also in the Sumaco volcano [2].

Several studies point to diminished metabolic activity with increased altitude. Jha et al. [21] found that altitude was a determinant of the relationship between microorganismal activity and metabolic activity. To test the potential effects of climate change on a Tibetan mountainous grassland, Yang et al. [22] sampled different elevations in a space-for-time substitution natural experiment, and they reported a higher abundance of cold-shock genes at higher elevations. Xu et al. [23] studied the physiological profiles at the community level of the soil microbiome along an elevational gradient in the Tibetan Plateau and found a significantly decreased richness and diversity of carbon source utilization at higher altitudes. Rofner et al. [24] found that climate-induced changes affect the structure and metabolic function of the soil bacterioplankton community in high-altitude ecosystems in Finland. Across tropical forests in the Andes, Nottingham et al. [25], evaluated long-term temperature effects and found increased microbial metabolic activity at higher temperatures. Along an elevational gradient, on the Tibetan Plateau, Feng et al. [26] evaluated the metabolic traits of the soil microbiome; they found that reduced metabolic efficiency was associated with investment in the acquisition of nutrients through enzymes. However, less definitive trends have also been reported. Smith et al. [27] used an elevation gradient as an analog of climate change and found that respiration and N-mineralization showed nonsignificant trends. Li et al. [28] sampled the soil microbiome along an elevational gradient in a Tibetan mountain grassland and found that soil carbon metabolism increased, independently of elevation, with temperature.

Building on our previous characterization of the Sumaco microbiome and its links to elevation and soil chemistry [2], we present an original and complementary effort to extract functional meaning from our 16S metabarcoding dataset. Here, we explore the comparative abundance patterns of inferred metabolic pathways from predicted functional profiling in a set of samples collected along elevation and sulfur gradients on the slopes of the Sumaco Volcano in Amazonian Ecuador. Based on the predicted abundance of functional genes, we provide an exploratory estimate of the inferred functional profiles of the bacterial community in contrast to two groups of samples divided by altitude or soil sulfur concentration (ppm). The latter is relevant for the unique geochemistry of this Amazonian volcano [2,4,29,30,31]. Ours is a data-driven exploration of the inferred functional landscape in Sumaco. Our study proposes establishing the essential foundation for future studies that aim to understand the significance of a unique volcanic geochemistry for microbial biodiversity in the Amazon region.

## 2. Materials and Methods

The dataset used for functional prediction was derived from the 16S rRNA gene amplicon sequences previously generated by the Ecuadorian Microbiome Project (EcuMP) from the Sumaco Volcano [2]. Briefly, total genomic DNA was extracted from soil samples using the PowerSoil DNA Isolation kit (MO BIO Laboratories, Inc., West Carlsbad, CA, USA). The V3–V4 hypervariable regions of the 16S rRNA gene were amplified and sequenced on an Illumina MiSeq platform (300 bp paired-end). Sequence data processing was performed using Mothur (v.1.43.0) following standard operating procedures, which included quality filtering, denoising, and chimera removal via VSEARCH. High-quality sequences were aligned to the SILVA v132 reference database and clustered into operational taxonomic units (OTUs) at 99% identity using the OptiClust algorithm. The resulting feature table and corresponding representative sequences were used as the source material for the current study.

We used the package PICRUSt2 (Phylogenetic Investigation of Communities by Reconstruction of Unobserved States) [32] to predict the functional potential of soil bacterial communities based on 16S rRNA gene sequences. The PICRUSt2 method uses a 41,926-record database of known microbial genomes, dereplicated to 20,000 16S rRNA gene clusters associated through a reference phylogenetic tree. To infer the functional capabilities of the sampled bacterial community, the prediction of pathway abundance is accomplished through a structured mapping of Enzyme Commission (EC) gene families to metabolic pathways. PICRUSt2 functional predictions are based on two molecular and metabolic pathway databases, MetaCyc and KEGG. We used the former database for its larger coverage of microorganisms and better pathway prediction performance [33]. The output of PICRUSt2 is a table of predicted gene families and their relative abundances (pathway abundances table), which can be used to compare the inferred functional potential of different microbial communities. We used the q2-PICRUSt2 plugin in QIIME2 for functional prediction. The plugin takes the feature table and the corresponding 16S rRNA sequences and generates a table of predicted gene families. The resulting output consisted of the predicted sample gene family profiles (EC and KO metagenome predictions) and predicted MetaCyc sample pathway abundances. The full set of instructions and applied algorithm is available at DOI: 10.5281/zenodo.16954911.

To reach a balance between comprehensibility and thoroughness, we consolidated the data table to Level 3 of the MetaCyc hierarchy, for a total of 149 inferred metabolic pathways. The MetaCyc reference table of hierarchical pathway ontologies was obtained from Supplementary Table S1 in ref. [34]. Thus, two data tables were necessary. The first with the MetaCyc pathway abundances, and the second with the metadata describing the differences in the samples (particularly their membership to two altitude groups and two sulfur concentration groups). The previously established statistical independence of altitude and sulfur, and their significance in explaining the main patterns of microbial diversity in the Sumaco volcano [2], allowed us to treat both variables separately. The altitude groups were divided into high and low depending on their presence above or below the median (2990 m), respectively. The sulfur groups were divided into high and low concentrations depending on the soil samples being above or below the median (4.44 ppm).

High-throughput sequencing (HTS) data is compositional data; thus, the total for measured abundances is arbitrary and determined by the sequencer machine [35]. Consequently, any statistical approach to the assessment of these kinds of data requires a specific strategy tailored for compositional data, such as the normalization of counts, within the framework of differential abundance analysis (DAA) [35]. Proper procedures for the analysis of compositional data produced by PICRUSt2 are possible by applying the tools implemented in ggpicrust2 [36].

For differential abundance analysis (DAA), we selected the DESeq2 method [37], which was implemented through the ggpicrust2 package. This choice was motivated by the robust statistical design of DESeq2 to handle overdispersed compositional and count data, a key characteristic of functional profiles derived from microbiome sequencing. The method utilizes a Negative Binomial model to accurately account for the variance structure of the data, employs a stable median-of-ratios normalization to correct for library size differences, and applies empirical Bayes shrinkage to stabilize the variance estimate [37]. This shrinkage procedure is particularly advantageous as it increases statistical power and reduces false positives by pooling information across all pathways, ensuring more reliable results. We applied a Benjamini–Hochberg (BH) correction to adjust p-values for multiple comparisons and to control the false discovery rate (FDR) of the statistical contrasts in the DAA analysis.

To adjust the proper behavior of the ggpicrust2 package for mapping the grouping variable in the metadata table across the samples in the data table, we implemented modifications to the source code of the pathway_errorbar.R file. This modification is available at the permanent repository with DOI: 10.5281/zenodo.16953153. For assistance in interpreting the significant inferred metabolic pathways, we used Gemini Advanced Deep Research with 2.5 Pro.

## 3. Results

The weighted average nearest sequenced taxon index (NSTI) for each sample, indicating the fit of ASVs and sequences to those in the reference MetaCyc pathway database, was 0.12 (SD = 0.03). While this value is within the range generally considered acceptable for soil environments [38], it also reflects a notable phylogenetic distance between the indigenous microbial taxa of the Sumaco volcano and the current reference genomes in the PICRUSt2 database. Consequently, these results should be interpreted as an exploratory estimation of functional potential rather than a definitive metagenomic census. An analysis of the inferred abundance table reveals a clear dominance of a few key metabolic pathways across the soil samples. The 10 most abundant inferred pathways, identified based on their overall predicted relative abundance, constitute a significant portion of the total metabolic potential (Figure 1).

The most prominent pathway predicted was Proteinogenic Amino Acid Biosynthesis, which consistently showed the highest predicted abundance in all samples, with a mean relative abundance of approximately 13.8%. This suggests its critical role as a core metabolic process in the microbial communities. Following the most predicted abundant pathway, purine nucleotide biosynthesis and fatty acid biosynthesis were inferred to have high relative predicted abundances, with mean values of 8.3% and 6.3%, respectively. Both inferred pathways exhibited some variability between samples, but they consistently ranked among the most dominant.

The remaining top 10 inferred pathways, including sugar biosynthesis, vitamin biosynthesis, and pyrimidine nucleotide biosynthesis, showed lower but still significant predicted abundances, with mean values ranging from 2.5% to 5.4%. These inferred pathways displayed a relatively consistent pattern in all samples (Figure 1), with their predicted minimum and maximum abundances not deviating widely from their mean, indicating a stable presence within the microbial communities.

In general, the inferred metabolic profile of the soil samples was characterized by predicted pathways related with fundamental anabolic processes, particularly those involved in the synthesis of amino acids, nucleotides, and lipids (Figure 1). Our estimates provide a general overview of the inferred pathways that were potentially present in the samples, highlighting those with the highest significant predicted abundance that may likely support core cellular functions within the microbial community.

### 3.1. Results of the Differential Abundance Analysis

Between the two altitude groups (Figure 2) and between the two sulfur concentration groups (Figure 3), the differential abundance analysis performed using DESeq2 on the microbial pathway data from soil samples revealed a set of inferred metabolic pathways that exhibited statistically significant differences in their predicted abundance. These pathways provide an inferred functional profile of the microbial communities and suggest how their metabolic potential varies between soil groups.

A total of 6 pathways, of a total of 149 pathways predicted by PICRUSt2, were found to be significantly different in their inferred functional gene abundance for the contrast between the two altitude groups (Figure 2 and Figure 4). Of a total of 149 pathways, 15 inferred metabolic pathways were significantly different in their predicted abundance in the contrast between low-sulfur and high-sulfur soils. A substantial number of inferred metabolic pathways had significantly higher predicted abundance in the high-sulfur soils compared to low-sulfur soils (Figure 3 and Figure 5). With the exception of one sample, clustering according to the inferred abundance of the significant pathways adhered well to predefined altitude or sulfur groups (Figure 3 and Figure 5). The discussion section presents an interpretation, based on currently available references for tropical and volcanic environments, of each of these significant pathways and their hypothetical potential role for the bacterial metabolic landscape of the Sumaco volcano.

#### 3.1.1. Pathways Enriched at High Altitude

Pathways related to fermentation (crotonate fermentation, DAA results: log_2_ fold change > 3; *p*-value < 0.001); cofactor biosynthesis (coenzyme B biosynthesis, DAA results: log_2_ fold change > 3; *p*-value < 0.001); sugar processing (sugar degradation, DAA results: log_2_ fold change < 1; *p*-value = 0.025); and hydrocarbon degradation (octante oxidation, DAA results: log_2_ fold change > 2; *p*-value = 0.001) were significant at higher altitudes (Figure 2).

#### 3.1.2. Pathways Enriched at Low Altitude

Two inferred pathways were significantly more abundant at low elevations; these were Gamma-aminobutyrate (GABA) degradation (or 4-aminobutanoate, DAA results: log_2_ fold change > −1; *p*-value = 0.025) and the biosynthesis of the osmoprotectant ectoine (DAA results: log_2_ fold change < −1; *p*-value = 0.025) (Figure 2).

#### 3.1.3. Pathways Enriched at High Sulfur

Under high-sulfur concentrations, the analysis suggested a higher predicted abundance of pathways associated with the degradation of complex organic carbon substrates, particularly aromatic compounds. These carbon substrates were toluene degradation (DAA results: log_2_ fold change > 1; *p*-value = 0.034); protocatechuate degradation (DAA results: log_2_ fold change < 1; *p*-value = 0.015); catechol degradation (DAA results: log_2_ fold change > 1; *p*-value = 0.046); 4-methylcatechol degradation (DAA results: log_2_ fold change > 3; *p*-value = 0.015); 3-phenylpropanoate degradation (DAA results: log_2_ fold change > 1; *p*-value = 0.035); syringate degradation (DAA results: log_2_ fold change > 2; *p*-value = 0.035); gallate degradation (DAA results: log_2_ fold change > 2; *p*-value = 0.035); aromatic biogenic amine degradation (DAA results: log_2_ fold change < 1; *p*-value = 0.035); and nitroaromatic compound degradation (DAA results: log_2_ fold change > 1; *p*-value = 0.045) (Figure 3).

Degradation pathways for other complex organic carbon substrates showed significantly higher predicted relative abundances in sulfur-enriched soils. Degradation associated with nitrogen metabolism was significantly represented by allantoin degradation (DAA results: log_2_ fold change < 1; *p*-value = 0.035), GABA (DAA results: log_2_ fold change < 1; *p*-value = 0.009), and proteinogenic amino acid degradation (DAA results: log_2_ fold change < 1; *p*-value = 0.046) (Figure 3).

Other significant pathways associated with the degradation of complex organic substrates were the detoxification processes of the superpathway of methylglyoxal degradation (DAA results: log_2_ fold change > 3; *p*-value < 0.001), nitroaromatic compound degradation (DAA results: log_2_ fold change > 1; *p*-value = 0.045), and anaerobic fermentation represented by butanediol biosynthesis (DAA results: log_2_ fold change > 3; *p*-value = 0.035) (Figure 3).

#### 3.1.4. Pathways Enriched at Low Sulfur

Of a total of 15 significant metabolic pathways associated with the two defined sulfur concentration groups, glycerol degradation (DAA results: log_2_ fold change < −3; *p*-value = 0.020) was the only pathway predicted to be significantly enriched in the low-sulfur samples. In contrast to those pathways significantly enriched in high-sulfur soils, which were related to the degradation of complex carbon molecules, this is a metabolic patwhay that is involved in the utilization of simpler carbon sources (Figure 3).

## 4. Discussion

### 4.1. Caveats and Challenges of This Study

We acknowledge the limitations inherent in our methodological approach while contextualizing the significance of our findings. The metabolic landscapes described in this study are based on functional predictions inferred from 16S rRNA gene data via PICRUSt2. As an inferential tool, its accuracy is dependent on the comprehensiveness of reference genome databases, and its predictions can have a weak association with the actual gene content present in the community (a limitation that may be particularly relevant in an underexplored ecosystem like the Sumaco volcano). We specifically acknowledge that this study lacks empirical validation through the quantitative PCR (qPCR) of key functional genes, such as those involved in sulfur cycling or direct enzymatic assays. Therefore, the metabolic shifts described herein represent testable hypotheses rather than confirmed physiological activities.

Furthermore, our current analysis provides a community-level overview of inferred functional potential but does not link these predicted functions back to specific taxa, thereby leaving an ecological perspective on the key microbial players unexplored. Recognizing these constraints, this study should be viewed as a foundational, data-driven exploration that generates critical hypotheses for future research. Subsequent efforts should prioritize shotgun metagenomics to validate these functional predictions and to connect metabolic pathways to the specific microbial lineages driving adaptation along Sumaco’s unique environmental gradients.

PICRUSt2 predictions are based on reference genomes from phylogenetically related taxa; thus, pioneering a microbiome exploration of a volcano in the Amazon region—such as Sumaco—is, at the same time, a privileged advantage, but also a potential weakness when bioinformatic tools, such as PICRUSt2, rely heavily on databases built mostly with data from higher latitudes. Genomic representation is evidently limited for tropical extremophilic environments, such as the Sumaco volcano. The absence of local indigenous species from global reference databases likely contributes to the observed NSTI values, potentially masking the unique metabolic adaptations specific to this Amazonian volcanic geochemistry. Our contribution may draw attention to this unique site and perhaps contribute to incentivize additional surveys to enrich genomic representation for these areas.

### 4.2. The Estimated Metabolic Core and the Influence of Sulfur

A salient feature of the Sumaco soil microbiome is the stability of the most abundant predicted pathways. The top ten pathways, which account for a significant proportion of the total metabolic potential, are dominated by essential anabolic routes, such as proteinogenic amino acid biosynthesis, purine nucleotide biosynthesis, and fatty acid biosynthesis. These pathways exhibit relatively low variation across samples, regardless of elevation or sulfur content. This pattern suggests the presence of a robust conserved metabolic core; these are a suite of fundamental housekeeping functions required for cellular maintenance and growth that remains stable despite environmental fluctuations. This functional redundancy likely ensures the resilience of the microbial community, allowing basic biological processes to persist, even as specific adaptive pathways shift in response to stress.

While altitude is a well-known driver of ecological zonation, our results indicate that soil sulfur concentration exerts a considerably stronger influence on the inferred functional profile of the Sumaco microbiome. We identified fifteen metabolic pathways that are significantly associated with sulfur levels compared to only six pathways linked to altitude, a finding that aligns with our previous assessment of microbial community structure, where sulfur explains more of the variation than elevation or pH [2]. This dominance of sulfur-driven functional differentiation underscores the unique geochemical nature of the Sumaco volcano, where intrinsic soil chemistry appears to override the classic climatic effects of the elevational gradient.

### 4.3. Pathways More Abundant at High Altitude

Pathways related to fermentation (crotonate fermentation), cofactor biosynthesis (coenzyme B biosynthesis), sugar processing (sugar degradation), and hydrocarbon degradation (octante oxidation) were predicted to be more abundant at higher altitudes. Although direct measurements of soil redox potential or moisture were not conducted in our study, we hypothesize that the inferred functional profile at high altitudes reflects an adaptation to environmental heterogeneity, where oxygen availability is chronically limited by high soil moisture, colder temperatures, and slow decomposition of complex organic matter [25,39,40].

#### 4.3.1. Crotonate Fermentation

An anaerobic metabolic pathway employed by certain bacteria to ferment crotonate, a four-carbon unsaturated fatty acid, into products such as acetate and other short-chain fatty acids [41,42]. The higher predicted abundance of crotonate fermentation at high altitudes is consistent with an adaptation to environments with limited oxygen availability. Such conditions can arise in montane soil due to several factors common at higher elevations: lower temperatures reducing oxygen diffusion rates, higher soil moisture content potentially leading to waterlogging (especially in cloud forest zones), and the presence of dense soil organic matter (SOM) creating anaerobic microhabitats [39,40]. Fermentation allows microbes to continue generating energy under these environmental constraints [43]. The specificity towards crotonate-related metabolism implies more than just a general capacity for anaerobiosis; it suggests a potential adaptation for utilizing the intermediate metabolites generated during the decomposition of organic matter that may be prevalent at high altitudes. High-elevation plant communities often differ significantly from lower elevations [44], and slower decomposition rates in colder conditions can lead to the accumulation of distinct organic compounds [25,39]. We hypothesize that microbes equipped for crotonate fermentation could possess a competitive advantage in processing these specific substrates under the prevailing anaerobic conditions.

#### 4.3.2. Coenzyme B

Related to the Vitamin B12 family of cofactors (cobalamins or related corrinoids) [45,46], and B12-dependent enzymes catalyze critical reactions, such as methyl group transfers, isomerizations, and reductions, often playing key roles in anaerobic respiration, fermentation, methanogenesis, the breakdown of complex organic molecules, and one-carbon (C1) metabolism [45,47,48]. The higher predicted abundance of coenzyme B biosynthesis (B12-dependent enzymes) is likely linked to the prevalence of metabolic strategies favored under high-altitude conditions, such as anaerobic/fermentative pathways [43] and the degradation of more complex or recalcitrant SOM that accumulates in colder environments [25,39]. The observation that biosynthesis pathways are predicted to be more abundant compared to uptake systems (where there are no significant predicted pathways of this kind) is particularly telling. It implies that either the external supply of B12 is limited at high altitudes, perhaps due to lower production rates by other microbes in the cold, instability in potentially acidic soils [39], or different community composition, or that the dominant high-altitude microbes have intrinsically higher requirements for B12 and possess the genetic capacity for self-sufficiency [46].

The higher predicted abundance of both anaerobic pathways, crotonate fermentation, and coenzyme B biosynthesis, in the high-altitude soil samples from the Sumaco volcano, could indicate the presence of more prevalent anaerobic microenvironments in these soils compared to lower elevations.

#### 4.3.3. Sugar Degradation

Higher predicted abundance of sugar degradation pathways suggests a community capacity for processing available carbohydrates, likely sourced from the abundant plant litter and root exudates associated with the high SOM content typical of higher elevations [39,40].

#### 4.3.4. Octane Oxidation

The factors driving the higher predicted abundance of octane oxidation at high altitudes on Sumaco warrant further investigation. Our study lacks specific data on soil hydrocarbon concentrations or detailed vegetation composition; thus, we can only hypothesize regarding the ecological drivers of this metabolic potential. We propose that these pathways may be involved in the degradation of naturally occurring long-chain alkanes, such as those found in plant cuticular waxes. It is plausible that the accumulation of these recalcitrant plant-derived compounds, potentially driven by slower decomposition rates at higher elevations, selects for microbial communities capable of alkane oxidation. However, confirming this hypothesis requires future studies that integrate soil metabolomics with botanical surveys to identify the specific substrates present.

### 4.4. Pathways with Higher Abundance at Low Altitude

#### 4.4.1. Gamma-Aminobutyrate (GABA) or 4-Aminobutanoate Degradation

A non-protein amino acid that plays diverse roles in various organisms. It serves as a potential carbon and nitrogen source for microbes [49]. The predicted higher abundance of GABA degradation pathways suggests that GABA is a more readily available or preferentially utilized C/N source for the microbial community at lower altitudes on Sumaco. This could stem from higher rates of GABA production by low-altitude vegetation, faster overall decomposition and nutrient turnover rates driven by warmer temperatures [25,39], releasing more labile compounds like GABA, or the dominance of specific microbial groups adept at its catabolism. The predicted higher abundance of GABA degradation pathways implies that microbes at lower elevations are geared towards rapidly exploiting relatively simple, N-containing organic molecules. This contrasts sharply with the high-altitude predicted abundance profile, which suggests a greater genomic potential for breaking down complex polymers (indicated indirectly by B12 dependence) and potentially slower nutrient cycling. This inferred functional difference may reflect adaptation to the distinct temperature regimes and consequent differences in the speed and nature of biogeochemical processes along the gradient.

#### 4.4.2. Ectoine Biosynthesis

An osmoprotectant (DAA results: log_2_ fold change < −1; *p*-value = 0.025) [50,51,52]. Under conditions of environmental stress, primarily high salinity or desiccation, ectoine contributes to maintain cellular turgor and water balance without interfering with metabolic processes [50,51,52]. The reasons behind potential osmotic stress in the low-elevation environment of the Sumaco volcano require evidence that is beyond the scope of our current study. In poorly understood ways that may significantly impact how bacteria in these soils have evolved adaptations to stress, the volcanic nature of Sumaco’s soils might also contribute to fluctuations in water retention and solute concentrations [53]. The inferred higher abundance of ectoine biosynthesis suggests that managing such abiotic stresses could be a key adaptive strategy for microbial survival and activity at the lower end of the Sumaco gradient. This highlights that the nature of environmental stress changes significantly with altitude, selecting for different protective mechanisms: osmoprotection at lower elevations versus adaptations to chronic cold, low O_2_ and recalcitrant substrates higher up.

### 4.5. Pathways More Abundant in High Sulfur

Sulfur is an indispensable macronutrient for both plants and microorganisms, participating in the synthesis of vital biomolecules, like cysteine, methionine, and various enzyme cofactors, thereby influencing a wide spectrum of metabolic pathways [54]. While pathways annotated as toluene or nitroaromatic degradation were inferred to be significantly enriched, we do not wish to imply the presence of these specific xenobiotics in the pristine Sumaco environment. Instead, we propose these inferred enriched pathways as functional indicators of a community equipped with versatile oxygenases and reductases. These enzymes likely participate in the broader carbon and nitrogen cycles by degrading naturally occurring, recalcitrant aromatic complexes derived from the dense Amazonian plant litter.

#### 4.5.1. Toluene Degradation

Previous studies have demonstrated a direct link between toluene degradation and sulfur cycling, with the enrichment of sulfate-reducing bacteria, including members of the *Desulfobulbaceae* family, being observed in association with this process, particularly anaerobic toluene degradation through sulfide-mediated electron transfer [55]. Toluene, or other aromatic compounds (e.g., plant waxes), potentially originate from plant litter or other sources, and they could then serve as a carbon source for microbial communities capable of coupling its breakdown to the reduction of the readily available sulfate [56].

#### 4.5.2. Catechol Degradation

A central intermediate in the microbial degradation of various aromatic compounds, including those derived from lignin, plant secondary metabolites, and anthropogenic pollutants [57]. Ortho cleavage for 4-methylcatechol is a further breakdown process [58,59]. Certain bacteria can utilize sulfur-containing aromatic compounds, such as sulfonates, as a source of sulfur for growth, and the degradation of these sulfonates could involve metabolic pathways that also process other aromatic compounds, which could lead to the formation of catechol as an intermediate [54]. Desulfonation metabolisms, particularly the desulfonation of benzenesulfonic acid, induce the formation of catechol, which is then oxidized by the ortho pathway [60]. The preference for the ortho-cleavage of 4-methylcatechol might also be associated with specific bacterial groups that are more competitive under certain sulfur regimes [59].

#### 4.5.3. Protocatechuate, Gallate, and Syringate Degradation Pathways

Another crucial intermediate in the degradation of various aromatic compounds, particularly those derived from lignin, which is a major component of plant cell walls [61,62,63,64]. Gallate and syringate are also phenolic compounds originating from the breakdown of lignin [62,63,65]. Sulfur plays an indirect but essential role in these pathways as it is a vital nutrient for plant growth and the synthesis of lignin-containing biomass [66,67]; however, we did not characterize the vegetation structure or litter composition in this study. Consequently, the relationship between the observed enrichment of lignin degradation pathways and soil sulfur levels remains speculative. We hypothesize that the high sulfur availability in these soils may indirectly favor these pathways by potentially alleviating nutrient limitations and promoting plant productivity [68], which could, in turn, increase the input of lignin-derived aromatics into the soil. While it is ecologically plausible that higher sulfur levels could support a larger community of lignin-degrading microbes [65], verifying this bottom-up trophic effect requires further analysis of the local plant biomass and soil organic matter composition.

#### 4.5.4. 3-Phenylpropanoate and 3-(3-Hydroxyphenyl) Propanoate Degradation

Aromatic acids derived from the breakdown of plant matter, including lignin and flavonoids [69]. Like other plant-derived aromatic degradation pathways, the predicted abundance of these pathways might be indirectly related to higher sulfur availability influencing plant growth, the composition of plant tissues, and the subsequent input of these specific aromatic compounds into the soil environment [66,67].

#### 4.5.5. Butanediol Biosynthesis

2,3-butanediol is a volatile bacterial compound produced by certain soil bacteria (rhizobacteria), and it has been shown to play a role in enhancing plant resistance to drought and inducing systemic resistance against pathogens, as well as contributing to bacterial fitness in the rhizosphere environment [70]. The butanediol biosynthesis pathway is predictably more abundant in the high-sulfur soils in Sumaco, and this could suggest that these conditions favor the microbial production of this compound, potentially as a stress tolerance mechanism. High-sulfur concentrations or specific sulfur compounds can be stressful to some organisms, and butanediol is known to act as a compatible solute and protects cells against various stresses [54].

#### 4.5.6. Superpathway of Methylglyoxal Degradation

The higher predicted abundance of the methylglyoxal (MG) degradation super pathway in the high-sulfur Sumaco soils is ecologically significant for several reasons. Firstly, a high-sulfur environment, arising from the redox cycling of sulfur compounds [71,72,73] and potentially exacerbated by the presence of redox-active metals [74,75], is inherently prone to oxidative stress. We hypothesize that this environment may promote oxidative stress, which could subsequently drive the endogenous production of MG. In this context, efficient detoxification via the glyoxalase pathway would represent a potentially advantageous trait [76,77]. Secondly, the dependence of the predominant glyoxalase system (GlyI/II) on Glutathione (GSH) establishes a direct biochemical link to sulfur metabolism. GSH is a cysteine-containing peptide [78,79,80], and cysteine biosynthesis represents the primary route for assimilating inorganic sulfur into organic molecules in many microorganisms [81]. Furthermore, we propose that the high sulfur availability might support elevated rates of cysteine synthesis and, consequently, higher intracellular pools of GSH. This enhanced GSH availability would, in turn, bolster the capacity and efficiency of the GSH-dependent glyoxalase system. Thus, the predicted higher abundance of MG degradation pathways may be associated with the requirements of sulfur-related oxidative stress (increasing MG levels) and could be facilitated by the high sulfur availability (supporting GSH synthesis).

#### 4.5.7. Allantoin Degradation

The higher predicted abundance of allantoin degradation pathways could be linked to environmental stress. The multiple stressors associated with the high-sulfur volcanic soil (sulfur toxicity, acidity, potential metal stress, and oxidative stress) [72,74,75] are conditions known to induce allantoin accumulation in plants [82,83,84,85]. Although we did not assess the plant stress markers in this study, we speculate that the predicted enrichment of allantoin degradation pathways may reflect a microbial capitalization on the allantoin released into the soil by stressed plant life. Alternatively, this pathway could be involved in endogenous microbial stress management.

#### 4.5.8. Gamma-Aminobutyrate (GABA) or 4-Aminobutanoate Degradation

A major physiological role of GABA metabolism, particularly in bacteria, is its contribution to acid stress tolerance; through the synthesis of GABA, intracellular protons are consumed in an irreversible decarboxylation reaction [86,87,88]. As a consequence, while GABA synthesis is key for immediate acid resistance, the subsequent degradation of accumulated GABA is necessary to regenerate glutamate (the substrate for GAD) back into central metabolism for energy production or biosynthesis [89]. The high sulfur content in the measured soil samples suggests the potential for localized acidification due to microbial sulfur oxidation, even within a potentially alkaline bulk soil matrix [72,73,90]. Although we did not quantify the oxidation rates or micro-scale pH variations, we propose a plausible mechanism, where localized acidification occurs within the soil micro-environment, even if the bulk soil matrix remains alkaline. In this scenario, the synthesis of GABA could serve as a bacterial strategy for coping with such acid stress. We hypothesize that the predicted enrichment of this degradation pathway reflects a community response to these geochemical conditions; however, confirming this link between sulfur oxidation, acidity, and GABA metabolism requires further physiological validation.

#### 4.5.9. Nitroaromatic Compound (NAC) Degradation

Nitroaromatic compounds are ecologically recalcitrant xenobiotics that are introduced into the environment primarily through human activities, including the use of pesticides, explosives, and industrial chemicals [91,92]. The predicted higher abundance of pathways annotated for nitroaromatic compound degradation in high-sulfur Sumaco soils is intriguing, given that nitroaromatic compounds are not often found in nature. Microbial degradation is a key process in their removal from contaminated soils [91,92]. A possible explanation for the higher predicted abundance of this pathway lies in the broad substrate specificity of the enzymes involved, particularly nitroreductases and potentially certain oxygenases [93,94]. The soil associated with the volcanic activity of Sumaco may potentially contain a variety of toxic organic molecules; thus, enzymes typically associated with NAC degradation, owing to their relaxed substrate specificity [95], may be recruited by the soil microorganisms to detoxify these other, more environmentally relevant toxic compounds, which may be imposed by the high-sulfur geochemistry.

#### 4.5.10. Proteinogenic Amino Acid Degradation

Encompasses the diverse metabolic pathways responsible for the catabolism of the 20 standard amino acids incorporated into proteins [96]. The high sulfur content of the Sumaco soils may specifically impact the metabolism of sulfur-containing amino acids (cysteine and methionine), with increased rates of sulfur assimilation [97,98,99]. Therefore, the higher predicted abundance of amino acid degradation pathways, particularly those involving cysteine and methionine, likely signifies not only a strategy for general nutrient acquisition, but also a specific adaptation to manage the high flux of sulfur through cellular metabolism.

### 4.6. Pathways with Higher Abundance at Low Sulfur

#### Glycerol Degradation

Glycerol is a simple sugar alcohol that can be produced in soil through various processes, including the breakdown of lipids and as a byproduct of certain microbial fermentation (Magalhães et al., 2024 [100]). Several soil microorganisms possess the ability to degrade glycerol under both aerobic and anaerobic conditions (Qatibi et al., 1991; Santos et al., 2018 [101,102]). Notably, studies have shown that glycerol can serve as an effective electron donor for sulfate reduction by sulfate-reducing bacteria (SRB) (Qatibi et al., 1991; Santos et al., 2018 [101,102]). In the presence of sulfate, SRB can oxidize glycerol, producing sulfide and other metabolic byproducts (Qatibi et al., 1991; Santos et al., 2018 [101,102]). Thus, sulfate-reducing bacteria, using glycerol as a carbon source for sulfate reduction, might be more active in high-sulfur soils.

## 5. Conclusions

Our study provides the first exploration of the predicted functional microbial landscape of Sumaco. There is a specific context for the Sumaco volcano: it has a unique alkaline geochemistry derived from back-arc magmatism, the high rainfall characteristic of the eastern Andean slopes, relatively undisturbed forest ecosystems, and soils derived from volcanic materials (which can influence properties like water retention and mineral nutrient supply [2,4,29,30,31]).

Our findings lead us to propose several hypotheses on how elevation and sulfur may be determining metabolic changes and adaptations on the microbiome in the Sumaco volcano. (1) We hypothesize that the high intrinsic sulfur content of Sumaco’s alkaline soils supports a robust community of sulfate-reducing bacteria that are the primary drivers of the anaerobic decomposition of complex plant-derived organic matter. (2) We hypothesize that the inferred functional differences observed along the elevational gradient reflect a fundamental trade-off in microbial life-history strategies, balancing rapid growth on simple substrates in warmer, low-altitude soils against specialized enzymatic capabilities for degrading complex polymers in colder, high-altitude soils. (3) We hypothesize that the higher predicted abundance of detoxification and stress–response pathways in both high-sulfur and low-altitude environments reflects distinct chemical and physical pressures, namely oxidative and toxic compound stress in high-sulfur soils, as well as osmotic stress in low-altitude soils.

Our study on the Sumaco volcano presents indirect evidence that high-altitude communities are predicted to possesses an enhanced potential for anaerobic metabolism (crotonate fermentation), synthesis of vital cofactors for complex reactions (coenzyme B12), and the degradation of diverse carbon sources, including sugars and hydrocarbons (sugar degradation and octane oxidation). We hypothesize that this inferred functional profile aligns with adaptation to the colder, wetter, and potentially oxygen-limited conditions, as well as the abundant, complex soil organic matter characteristic of high tropical elevations [25,39,40], thereby providing a testable framework for future studies incorporating direct soil sensor data for soil moisture or redox potential. The higher predicted abundance of pathways typically associated with both anaerobic and aerobic processes underscores the importance of environmental heterogeneity and metabolic flexibility in these environments. This high-altitude environment. Conversely, low-altitude communities show increased predicted abundance for degrading specific labile nitrogenous compounds (GABA degradation) and for synthesizing protective compatible solutes (ectoine biosynthesis). This suggests adaptation to warmer conditions is likely to support faster nutrient cycling, but that it could also impose significant abiotic stresses, the latter of which has not been identified yet.

The inferred functional differences between high-sulfur and low-sulfur soil microbial communities are best understood within the unique biogeochemical context of the Sumaco Volcano. Unlike typical Andean volcanoes producing more acidic materials, Sumaco is characterized by its distinct alkaline lavas (basanite to tephriphonolite), a consequence of its unique geological setting over 100 km east of the main Andean arc front [2]. Petrological studies indicate these magmas were likely sulfate-saturated during differentiation at shallow crustal levels, leading to high intrinsic sulfur concentrations in the parent material, as evidenced by the presence of the sulfate-bearing mineral haüyne [4,29,31].

The unique alkaline geochemistry of Sumaco [2,30,31] may exert a significant influence. Although sulfur oxidation processes typically lead to soil acidification [103,104], the inherent buffering capacity of alkaline minerals could moderate pH changes, potentially maintaining near-neutral conditions despite active sulfur cycling. This might explain why sulfur concentration, rather than pH (both uncorrelated), emerged as the most informative physicochemical parameter structuring bacterial communities in our previous study at Sumaco [2]. Sulfur availability, particularly as sulfate acting as an electron acceptor, may be the primary ecological filter in this system. The high sulfate levels, combined with ample organic matter in the tropical mountain setting [25,39], may strongly favor anaerobic respiration via sulfate reduction, linking sulfur availability directly to the higher predicted abundance of the complex organic matter degradation observed in the high-sulfur samples.

The enhanced degradation of plant litter releases a variety of secondary metabolites, including potentially toxic or inhibitory phenolics, tannins, and alkaloids, some of which contain nitro groups [105]. Methylglyoxal detoxification might have higher predicted abundance due to the metabolic shifts associated with anaerobic conditions or specific substrate utilization patterns [106]. Thus, the predicted higher abundance of detoxification pathways likely reflect a community equipped to handle the chemical challenges inherent in a high-sulfur, high-decomposition environment. The concurrent predicted higher abundance of pathways degrading nitrogenous compounds—allantoin, GABA, proteinogenic amino acids, and aromatic biogenic amines—suggests that nitrogen cycling may be tightly interwoven with carbon and sulfur metabolism in high-sulfur soils. Finally, the higher predicted abundance of butanediol biosynthesis signifies the potential for fermentation to occur alongside anaerobic respiration. Anaerobic environments are often heterogeneous, and fermentation can serve as a crucial alternative or complementary strategy for energy generation and redox balancing (NAD+ regeneration) when sulfate becomes locally depleted or for specific substrates preferentially channeled through this route [43].

The inferred functional profile of low-sulfur soil bacterial communities contrasts sharply with the higher predicted abundance of glycerol degradation. Glycerol, a simple C3 substrate derived from lipid breakdown, feeds readily into central metabolic pathways like glycolysis. This shift towards simpler substrate utilization likely reflects a fundamental change in ecological strategy driven by sulfur limitation; however, a thorough explanation requires further exploration of the metabolic landscape within Sumaco volcanic soils.

Validation of the three hypotheses we have proposed will require a multi-pronged approach. Shotgun metagenomic sequencing is essential to confirm the presence and abundance of the predicted metabolic genes and to, crucially, link these inferred functions to the specific microbial taxa driving these processes. Furthermore, microcosm experiments manipulating temperature, moisture, and sulfate availability would allow for direct testing of these functional predictions, while cultivation and isolation of key organisms from Sumaco would provide the genomic resources needed to improve the accuracy of future bioinformatic inferences in this and other under-explored tropical volcanic ecosystems.

## Figures and Tables

**Figure 1 microorganisms-14-00094-f001:**
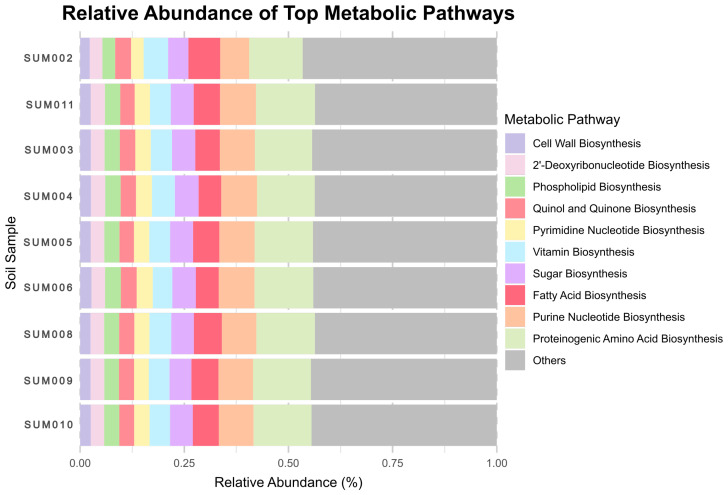
Predicted relative abundance of the inferred metabolic pathways across the soil samples from the Sumaco volcano. The first ten most abundant pathways are depicted in detail.

**Figure 2 microorganisms-14-00094-f002:**
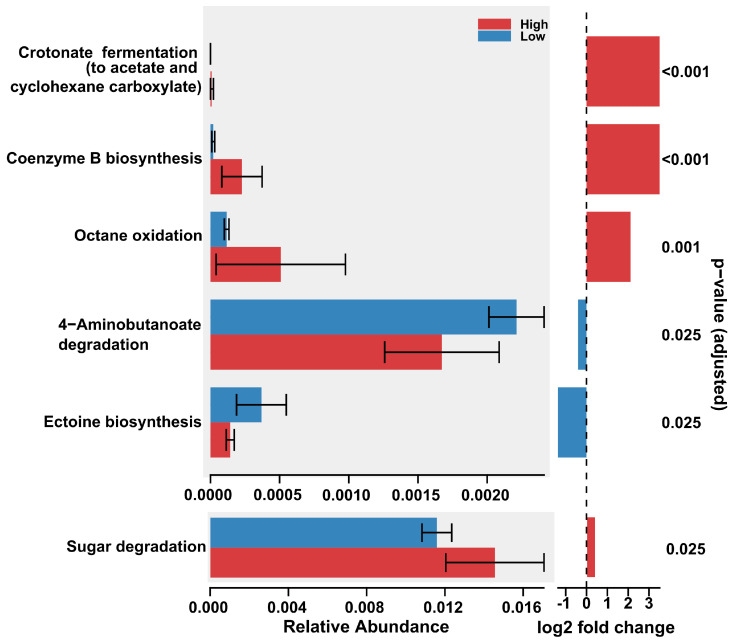
The results of a differential abundance analysis between altitude groups. Only significant pathways are depicted. The Benjamini–Hochberg (BH) correction was applied on the *p*-values to control for the false discovery rate.

**Figure 3 microorganisms-14-00094-f003:**
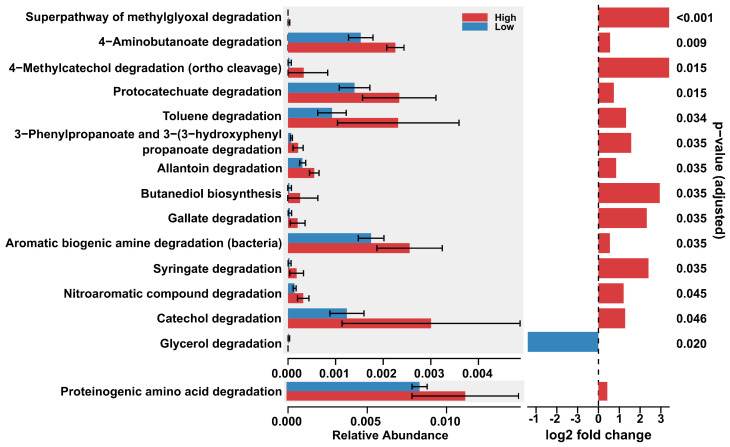
The results of a differential abundance analysis between sulfur groups. Only significant pathways are depicted. The Benjamini–Hochberg (BH) correction was applied on the *p*-values to control for the false discovery rate.

**Figure 4 microorganisms-14-00094-f004:**
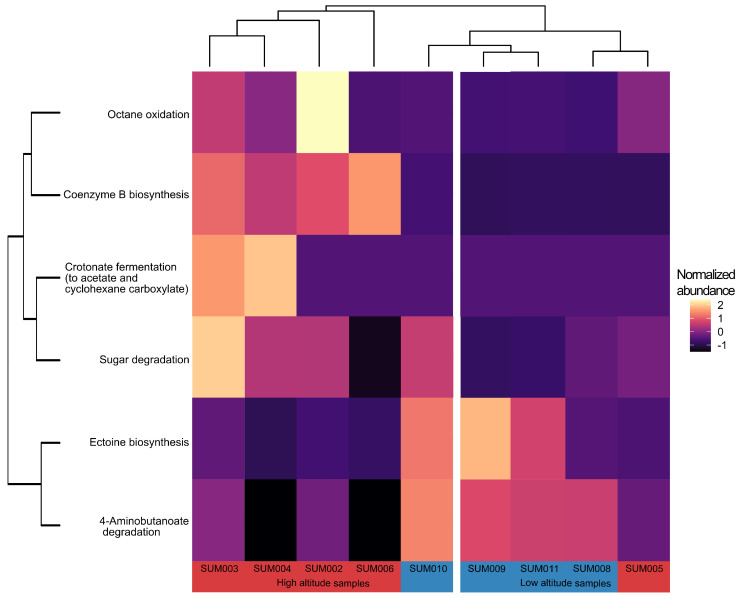
Heatmap depicting the relative predicted abundance of the significant pathways between altitude groups, as established by the differential abundance analysis.

**Figure 5 microorganisms-14-00094-f005:**
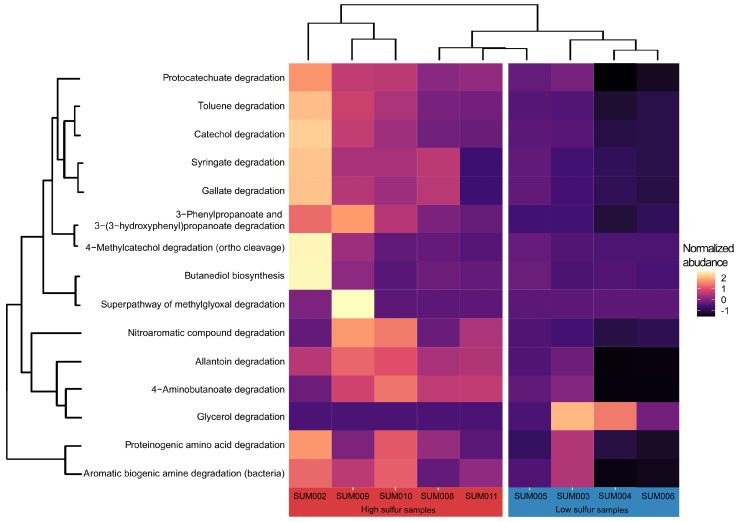
Heatmap depicting the relative predicted abundance of the significant pathways between sulfur groups, as established by the differential abundance analysis.

## Data Availability

The original data presented in this study are openly available in NCBI BioProject database under accession number (PRJNA744540).

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
