# Peer review of "Predicted Bacterial Metabolic Landscapes of the Sumaco Volcano: A Picrust2 Analysis of 16S rRNA Data from Amazonian Ecuador"

_microorganisms, 2026, doi:10.3390/microorganisms14010094_

Round 1
Reviewer 1 Report
Comments and Suggestions for Authors
The authors report an analysis of soil microbiomes from the largely undisturbed environments of the Sumaco volcano. Using 16S rRNA gene metabarcoding and predictive functional profiling, they evaluated how microbial metabolic potential varies along elevation and sulfur gradients. The results indicate that high-altitude communities show a higher predicted abundance of pathways associated with anaerobic metabolism, coenzyme B₁₂ synthesis, and the degradation of diverse carbon substrates. In contrast, sulfur-rich soils exhibit enriched pathways related to the decomposition of complex organic compounds and nitrogen metabolism, reflecting adaptations to Sumaco’s unique geochemistry. Although functional predictions have inherent limitations, the observed patterns suggest distinct ecological strategies shaped by the environmental heterogeneity of the volcano. This study highlights how geochemical gradients structure microbiome functional diversity and opens avenues for future research linking predicted functions to the taxa that encode them.
Overall, the Discussion aligns well with the Results, but some interpretations go beyond the available evidence and should be adjusted. One example concerns the proposed origin of hydrocarbons associated with the octane oxidationpathway, whose predicted abundance increases with elevation. The Discussion suggests that these compounds may originate from specific plant sources or other natural inputs; however, the study does not include data on hydrocarbon concentrations in the soil or on the composition of the local vegetation. Therefore, although plausible, this explanation should be clearly framed as a hypothesis rather than a conclusion supported by the data.
A similar issue arises in the proposed link between lignin input and sulfur availability. The Discussion posits that sulfur-rich soils may promote greater plant growth and consequently higher lignin deposition. While this is ecologically reasonable, no measurements of vegetation, biomass, or litter composition were collected. Thus, this interpretation should also be presented as speculative rather than as a direct inference from the inferred metagenomic profiles.
Another point requiring clarification concerns the ecological interpretation of the GABA pathway in sulfur-rich soils. The Discussion attributes its higher predicted abundance to acid-stress responses associated with sulfur oxidation. Although consistent with known physiology of sulfur-metabolizing microorganisms, the study does not include measurements of pH, sulfur species, or oxidation rates. As such, this explanation should be described as a plausible mechanism, not as a demonstrated process.
Conversely, several important findings are underemphasized in the Discussion. One is the strong stability of the ten most abundant pathways, which represent essential anabolic routes and show little variation across samples. This pattern suggests the presence of a conserved metabolic core in Sumaco soils, an aspect that merits explicit discussion. Additionally, the Results clearly show that sulfur exerts a stronger influence on the functional profile than elevation, with fifteen sulfur-associated pathways compared to only six linked to altitude. This difference, which identifies sulfur as a more influential environmental driver, should be highlighted more prominently.
In summary, the Discussion is scientifically sound and consistent with expected microbial ecology and biogeochemistry for these environments. However, it requires key clarifications to ensure that interpretations do not exceed the support provided by the data. Speculative explanations should be flagged as such, and central findings, such as the stable metabolic core and the stronger influence of sulfur relative to elevation, should receive greater emphasis to strengthen alignment between the Results and the Discussion.
En síntesis, la Discusión es sólida y coherente con lo que se espera de la ecología microbiana y la biogeoquímica de estos ambientes. Sin embargo, requiere aclaraciones clave para garantizar que ninguna interpretación exceda la evidencia disponible. Las explicaciones especulativas deben señalarse como tales, y los hallazgos centrales, como el núcleo metabólico estable y la mayor influencia del azufre respecto de la altitud, deben recibir un tratamiento más destacado para fortalecer la concordancia entre los Resultados y la Discusión.
Minor (Please review the English across the whole document).
Several other sentences need correction as well; some examples are provided below.
Line 26. “study by the Ecuadorian Microbiome Project (EcuMP) [2] on the Sumaco Volcano assessed the effects…” “Previous work on the Sumaco volcano examined how elevational and physicochemical gradients influence soil microbial richness and community structure [2].”
Line 32. “being this last not signifiantly correlated with neither of the former.” “and sulfur did not show significant correlation with either variable.”
Line 58. “and from the Sumaco volcano [2].” “also in the Sumaco volcano [2].”
Line 63. “they found that cold-shock genes were more abundant at higher elevations.” “and reported higher abundance of cold-shock genes at higher elevations.”
Line 69. “tested whether the growth responses… they found…” “evaluated long-term temperature effects and found increased microbial metabolic activity at higher temperatures.”
Line 80. “In a pioneering first effort in Ecuador…” “Building on our previous characterization of the Sumaco microbiome and its links to elevation and soil chemistry [2],”
Author Response
Comments 1: The authors report an analysis of soil microbiomes from the largely undisturbed environments of the Sumaco volcano. Using 16S rRNA gene metabarcoding and predictive functional profiling, they evaluated how microbial metabolic potential varies along elevation and sulfur gradients. The results indicate that high-altitude communities show a higher predicted abundance of pathways associated with anaerobic metabolism, coenzyme B₁₂ synthesis, and the degradation of diverse carbon substrates. In contrast, sulfur-rich soils exhibit enriched pathways related to the decomposition of complex organic compounds and nitrogen metabolism, reflecting adaptations to Sumaco’s unique geochemistry. Although functional predictions have inherent limitations, the observed patterns suggest distinct ecological strategies shaped by the environmental heterogeneity of the volcano. This study highlights how geochemical gradients structure microbiome functional diversity and opens avenues for future research linking predicted functions to the taxa that encode them.
Response 1: We are grateful to Referee 1 for the thorough review and constructive feedback. We have carefully followed all the suggestions and corrections proposed by the referee so we can improve the quality of our study. To facilitate tracking the improvements made to our manuscript, please refer to text highlighted in yellow.
Comments 2: Overall, the Discussion aligns well with the Results, but some interpretations go beyond the available evidence and should be adjusted. One example concerns the proposed origin of hydrocarbons associated with the octane oxidation pathway, whose predicted abundance increases with elevation. The Discussion suggests that these compounds may originate from specific plant sources or other natural inputs; however, the study does not include data on hydrocarbon concentrations in the soil or on the composition of the local vegetation. Therefore, although plausible, this explanation should be clearly framed as a hypothesis rather than a conclusion supported by the data.
Response 2: We agree with the referee that we lacked the chemical and botanical data to definitively attribute the octane oxidation potential to specific plant sources. We have rewritten subsection "Octane oxidation" (in Section 4.3) to explicitly state that we did not measure hydrocarbon concentrations or vegetation composition. We have rephrased the interpretation to propose this as a hypothesis.
Comments 3: A similar issue arises in the proposed link between lignin input and sulfur availability. The Discussion posits that sulfur-rich soils may promote greater plant growth and consequently higher lignin deposition. While this is ecologically reasonable, no measurements of vegetation, biomass, or litter composition were collected. Thus, this interpretation should also be presented as speculative rather than as a direct inference from the inferred metagenomic profiles.
Response 3: We appreciate this correction. We have modified the text in the Discussion (Subsection 4.5) regarding "Protocatechuate, gallate and syringate degradation pathways." We now explicitly acknowledge that we did not characterize the vegetation or litter and characterize the link between sulfur availability and lignin input as a speculative, "bottom-up" trophic hypothesis.
Comments 4: Another point requiring clarification concerns the ecological interpretation of the GABA pathway in sulfur-rich soils. The Discussion attributes its higher predicted abundance to acid-stress responses associated with sulfur oxidation. Although consistent with known physiology of sulfur-metabolizing microorganisms, the study does not include measurements of pH, sulfur species, or oxidation rates. As such, this explanation should be described as a plausible mechanism, not as a demonstrated process.
Response 4: The referee is correct that without micro-scale pH measurements or oxidation rates, we cannot definitively claim acid stress is occurring. We have revised the subsubsection on "Gamma-aminobutyrate (GABA) or 4-aminobutanoate degradation" in Section 4.5. to clarify that this is a proposed mechanism based on potential micro-environmental conditions rather than a demonstrated physiological process.
Comments 5: Conversely, several important findings are underemphasized in the Discussion. One is the strong stability of the ten most abundant pathways, which represent essential anabolic routes and show little variation across samples. This pattern suggests the presence of a conserved metabolic core in Sumaco soils, an aspect that merits explicit discussion. Additionally, the Results clearly show that sulfur exerts a stronger influence on the functional profile than elevation, with fifteen sulfur-associated pathways compared to only six linked to altitude. This difference, which identifies sulfur as a more influential environmental driver, should be highlighted more prominently.
Response 5: We thank the referee for identifying these underemphasized points. We have inserted a new section ("Section 4.2: The Conserved Metabolic Core and the Dominant Influence of Sulfur") to explicitly discuss the stability of anabolic pathways and to quantitatively highlight the stronger influence of sulfur (15 significant pathways) compared to altitude (6 pathways).
Comments 6: In summary, the Discussion is scientifically sound and consistent with expected microbial ecology and biogeochemistry for these environments. However, it requires key clarifications to ensure that interpretations do not exceed the support provided by the data. Speculative explanations should be flagged as such, and central findings, such as the stable metabolic core and the stronger influence of sulfur relative to elevation, should receive greater emphasis to strengthen alignment between the Results and the Discussion.
Response 6: To ensure consistency with the referee's guidance regarding ecological interpretation, we proactively reviewed the remainder of the Discussion and applied similar adjustments to other sections that relied on unmeasured variables (specifically the subsubsections "Crotonate Fermentation", "Superpathway of methylglyoxal degradation", and "Allantoin degradation"). We have softened the language in these sections to frame the interpretations as hypotheses rather than definitive conclusions.
Comments 7: En síntesis, la Discusión es sólida y coherente con lo que se espera de la ecología microbiana y la biogeoquímica de estos ambientes. Sin embargo, requiere aclaraciones clave para garantizar que ninguna interpretación exceda la evidencia disponible. Las explicaciones especulativas deben señalarse como tales, y los hallazgos centrales, como el núcleo metabólico estable y la mayor influencia del azufre respecto de la altitud, deben recibir un tratamiento más destacado para fortalecer la concordancia entre los Resultados y la Discusión.
Response 7: We have carefully reviewed our work to comply with your valuable comments.
Comments 8: Minor (Please review the English across the whole document).
Response 8: We thank the reviewer for this observation. We have conducted a comprehensive linguistic review of the manuscript to improve clarity, grammar, and technical precision. Specific corrections include:
1. LINE 10: We improved sentence structure. We changed "...inferentially associated with..." to "...associated with an inferred enrichment of...". [This was the only change we could not highlight in yellow due to limitations in the LaTeX highlightintg package for the abstract section]
2. LINE 110: We fixed a typo for the word "Commission".
3. LINE 387: We fixed an error in verb conjugation. We changed "The reasons behind potential osmotic stress... requires evidence that is beyond the reach of our current study." with "The reasons behind potential osmotic stress in the low elevation environment of the Sumaco volcano require evidence that is beyond the scope of our current study."
4. LINE 544: We improved sentence structure. We changed "Our findings invite proposing a few hypotheses..." with "Our findings lead us to propose several hypotheses...".
5. LINE 611: We improved sentence structure. We changed "...driven by sulfur limitation, but a thorough explanation will require further exploration of the metabolic landscape in the soils of the Sumaco volcano." with "...driven by sulfur limitation; however, a thorough explanation requires further exploration of the metabolic landscape within Sumaco volcanic soils.".
Comments 9: "Several other sentences need correction as well; some examples are provided below..."
Response 9: As suggested by the referee, we have acknowledged all improvements to the specified sentences. For the first sentence pointed out by the referee (LINE 26) we have chosen a different improvement than the suggested. We also applied a variation to the suggested improvement for the sentence at LINE 32. As the line numbering has been readjusted, please refer to lines 58, 63, 69, and 79.
We extend our gratitude to Reviewer 1 for the time invested in our manuscript and the insightful suggestions and corrections provided that have contributed to improving our work.
Reviewer 2 Report
Comments and Suggestions for Authors
This study explores the predicted functional profiles of soil bacterial communities in the Sumaco volcano using PICRUSt2 based on 16S rRNA amplicon data. The authors analyze the differential abundance of metabolic pathways across elevation and sulfur gradients. The topic is interseting, but there are questions to be addressed.
Main Comments:
1. The entire study relies on PICRUSt2 predictions. Given that the Amazonian volcanic soil is an under-explored ecosystem, the reference genome database likely lacks representation of local indigenous species. The NSTI value of 0.12 indicates a considerable phylogenetic distance from reference genomes. The authors must acknowledge this limitation more transparently. Ideally, qPCR validation of key functional genes (e.g., dsrA for sulfur metabolism) or enzymatic assays should be included. If not possible, the language must be softened throughout the manuscript (from "demonstrated" to "inferred" or "predicted").
2. The discussion heavily focuses on specific xenobiotic degradation pathways (e.g., toluene, octane, nitroaromatic compounds). These pathways are often subject to horizontal gene transfer and are notoriously difficult to predict accurately using 16S marker genes alone. Avoid over-interpreting these specific catabolic pathways unless there is chemical evidence of these substrates in the soil. Focus on core metabolic functions (C, N, S cycling) which are more reliably predicted.
3. The authors categorized continuous variables (Elevation and Sulfur concentration) into binary groups (High vs. Low). This approach leads to information loss.Since elevation and sulfur are continuous variables, a regression-based analysis (e.g., using MaAsLin2) would be more powerful and appropriate to identify trends than a simple binary comparison.
Minor Comments:
1. The title should reflect the predictive nature of the study. I suggest changing it to: "Predicted bacterial metabolic landscapes..." or "Inferred functional profiles...".
2. The heatmaps (Figures 4 & 5) should include clustering dendrograms to show sample similarity.
3. Please provide more context on why "anaerobic" pathways were predicted at high altitudes. Are there measurements of soil moisture or redox potential to support the hypothesis of anoxic conditions?
Author Response
Comments 1: This study explores the predicted functional profiles of soil bacterial communities in the Sumaco volcano using PICRUSt2 based on 16S rRNA amplicon data. The authors analyze the differential abundance of metabolic pathways across elevation and sulfur gradients. The topic is interseting, but there are questions to be addressed.
Response 2: We want to thank Reveiwer 2 for the time dedicated to our manuscript and for the constructive criticism that has allowed us to adjust and clarify the informaton presented on our work.
Comments 2: The entire study relies on PICRUSt2 predictions. Given that the Amazonian volcanic soil is an under-explored ecosystem, the reference genome database likely lacks representation of local indigenous species. The NSTI value of 0.12 indicates a considerable phylogenetic distance from reference genomes. The authors must acknowledge this limitation more transparently. Ideally, qPCR validation of key functional genes (e.g., dsrA for sulfur metabolism) or enzymatic assays should be included. If not possible, the language must be softened throughout the manuscript (from "demonstrated" to "inferred" or "predicted").
Response 2: We acknowledge the limitation inherent to our study by adding context to the measured NSTI value (lines 278-280). However, please notice that the strength of our study is the uniqueness of the Amazon biodiversity, particularly the remote and unexplored Sumaco volcano in the middle of the tropical forest; thus, the NSTI value in our study is a byproduct of the gap in knowlege for tropical microbiomes, that we are contributing to close. Given the lack of empirical validation, we have appropriately gauged our inferences on the microbiome by softening causal language. Please refer to lines 161 to 165 for an acknowledgement on the NSTI value in the context of our study. The lines 260 to 262, section "Caveats and challenges of the study" state the limitation of not performing qPCR and direct enzymatic assays. On this same section we stress the limited coverage of global reference databases for environments such as the Sumaco Volcano. We have changed the use of words in sentences or paragraphs where we propose possible metabolic ecological mechanisms, please refer to the following lines: 174, 222, 306-310, 323-325, 464-467, 481-485, 558, 561.
Comments 3: The discussion heavily focuses on specific xenobiotic degradation pathways (e.g., toluene, octane, nitroaromatic compounds). These pathways are often subject to horizontal gene transfer and are notoriously difficult to predict accurately using 16S marker genes alone. Avoid over-interpreting these specific catabolic pathways unless there is chemical evidence of these substrates in the soil. Focus on core metabolic functions (C, N, S cycling) which are more reliably predicted.
Response 3: We agree that without direct chemical evidence, 'xenobiotic' annotations should be interpreted with caution. However, many of these enzymes (e.g., oxygenases) possess relaxed substrate specificity and likely act on naturally occurring recalcitrant aromatic compounds from plant litter rather than industrial pollutants. We have adjusted the text (lines 401–407) to emphasize that these pathways are likely proxies for carbon and nitrogen mineralization. By grouping these results under core biogeochemical cycling, we maintain the metabolic insights.
Comments 4: The authors categorized continuous variables (Elevation and Sulfur concentration) into binary groups (High vs. Low). This approach leads to information loss. Since elevation and sulfur are continuous variables, a regression-based analysis (e.g., using MaAsLin2) would be more powerful and appropriate to identify trends than a simple binary comparison.
Response 4: We recognize that the categorization of continuous variables, such as elevation and sulfur concentration, into binary groups involves a trade-off between statistical power and biological interpretability. While regression-based models can identify linear trends, our study specifically aimed to characterize the contrasting metabolic landscapes that emerge at the ecological extremes of these gradients. This binary approach, supported by the statistical independence of both factors in our previous study (Díaz et al., 2022) (lines 32-33), allowed us to implement the robust DESeq2 framework, which is designed for high-sensitivity differential abundance analysis in compositional datasets. This approach allowed us to pinpoint pathways that are significantly enriched in a contrast between groups of samples in contrasting sulfur or altitude groups. Given the exploratory nature of this first microbiome survey of the Sumaco volcano, defining these discrete groups provided a clearer foundation for establishing our initial ecological hypotheses regarding the functional trade-offs and adaptive strategies within the community. References: Díaz, M.; Quiroz-Moreno, C.; Jarrín-V, P.; Piquer-Esteban, S.; Monfort-Lanzas, P.; Rivadeneira, E.; Castillejo, P.; Arnau, V.; Díaz, 637 W.; Sangari, F.J.; et al. Soil Bacterial Community Along an Altitudinal Gradient in the Sumaco, a Stratovolcano in the Amazon 638 Region. Frontiers in Forests and Global Change 2022, 5, 738568. https://doi.org/10.3389/ffgc.2022.738568.
Comments 5: Minor Comments:
1. The title should reflect the predictive nature of the study. I suggest changing it to: "Predicted bacterial metabolic landscapes..." or "Inferred functional profiles...".
2. The heatmaps (Figures 4 & 5) should include clustering dendrograms to show sample similarity.
3. Please provide more context on why "anaerobic" pathways were predicted at high altitudes. Are there measurements of soil moisture or redox potential to support the hypothesis of anoxic conditions?
Response 5: 1) We have changed the title of the manuscript accordingly; 2) We have added clustering dendrograms to Figures 4 and 5 and described the clustering pattern in lines 203 to 205; 3) We have expanded the discussion (lines 306–311) to provide the ecological context for predicted anaerobic pathways at high altitudes. While we did not measure redox potential directly, the combination of high rainfall (leading to waterlogging in cloud forest zones) and the accumulation of dense soil organic matter (SOM) creates a high likelihood of anoxic microenvironments where fermentation and B12-dependent metabolism become competitive strategies.
Round 2
Reviewer 1 Report
Comments and Suggestions for Authors
The authors have incorporated all the suggestions, and I consider that the manuscript is now suitable for publication